# Anatomical Morphology Analysis of Internal Jugular Veins and Factors Affecting Internal Jugular Vein Size

**DOI:** 10.3390/medicina56030135

**Published:** 2020-03-18

**Authors:** Jae Cheon Jeon, Woo Ik Choi, Jae Ho Lee, Sang Hun Lee

**Affiliations:** 1Department of Emergency Medicine, Dongsan Medical Center, Keimyung University School of Medicine, Daegu 42601, Korea; 2Department of Anatomy, Keimyung University School of Medicine, Daegu 42601, Korea

**Keywords:** jugular veins, central venous catheters, computed tomography, X-ray, age, body mass index

## Abstract

*Background and objectives:* There is a paucity of research on the shape of internal jugular vein (IJV) and their association with an individual’s morphology and various chronic diseases. Therefore, this study aimed to analyze the anatomy of the IJV across various patients and to relate the differences in anatomy to basic patient characteristics. *Materials and Methods:* This retrospective study included a total of 313 patients who underwent contrast-enhanced neck computed tomography between January 2017 and December 2018. The circumferences of the right and left IJVs were measured at three locations (hyoid bone, cricoid cartilage, and first thoracic vertebra) and parameters affecting the size of the IJV were analyzed. *Results:* The right IJV was significantly larger than the left IJV at each position (*p* < 0.001), and the area of the lumen was the largest at the cricoid cartilage level (*p* < 0.001). After dividing the right IJV data into two groups (above and below the median area), multivariate logistic regression analysis showed that age (odds ratio (OR) 1.040; 95% confidence interval (CI) 1.022–1.058, *p* < 0.001) and body mass index (BMI, OR 1.080; 95% CI 1.011–1.154, *p* = 0.0.23) affected size. *Conclusions:* The right IJV is larger than the left and has a rhomboid morphology. Age and BMI are significant factors affecting the IJV size.

## 1. Introduction

A large number of patients who visit the emergency room have their blood collected and catheters inserted via their peripheral blood vessels [1]. However, insertion of a central venous catheter can be performed in situations where peripheral blood vessels are difficult to access and where hyperosmolar or vasoconstrictive agents are used [2]. Central vein insertion can cause mechanical complications, such as arterial injury, hematoma, pneumothorax, and hemothorax, so care must be taken during the procedure [3]. Blood vessels used for central venous catheters include the jugular, subclavian, and femoral veins [3]. Among them, the internal jugular vein (IJV) is the most preferred because it is easy to access with ultrasound and has the fewest mechanical complications [3,4]. 

It is known that the right IJV is larger than the left, with a conical shape that widens caudally [5]. The relationships between the IJV and various central nervous system diseases, such as multiple sclerosis, Meniere’s disease, Alzheimer’s disease, and Parkinson’s disease, have been studied [6,7,8,9]. However, there is still a paucity of research on the shape of IJVs and their association with an individual’s morphology and various chronic diseases such as hypertension, diabetes, and stroke.

In this study, we looked at the anatomical shape of the IJVs of patients who underwent computed tomography (CT) of the neck and examined the differences in IJV size and shape according to individual characteristics.

## 2. Materials and Methods

### 2.1. Study Design and Search Strategy

We followed the STROBE (Strengthening the reporting of observational studies in epidemiology guidelines). This retrospective study included adult patients (aged ≥18 years) who underwent contrast-enhanced neck CT in the Dongsan Hospital of Daegu, Korea, from January 2017 to December 2018. The study protocol was approved by the institutional review board of Keimyung University Dongsan Hospital (no. 2019-08-060, data of approval, august 27, 2019), which waived the requirement for informed consent due to the retrospective nature of this study. In patients with stable vital signs, the neck CT scan was performed in the supine position with maintenance of full inspiration. Patients who had a mass in the neck, including the oral cavity, tonsil, nasopharynx, or thyroid, or who had a history of neck area surgery were excluded. Data lacking basic patient records required for the study such as height and weight were also excluded (Figure 1).

The maximal circumference and diameter were measured for the left IJV and measurements were taken on the right IJV at three different levels: The hyoid bone (upper), cricoid cartilage (middle), and the first thoracic vertebra (lower). The reason for choosing these landmarks was that these are frequently used by radiologists to make measurements [5].

Patient characteristics, including height, weight, age, sex, and previous illness history (including hypertension, diabetes, cardiovascular disease, cerebrovascular disease, and history of chemotherapy) were collected via electronic medical records. Cardiovascular diseases include diseases of the heart itself, such as angina, myocardial infarction, heart failure, and cardiac arrhythmias, as well as vascular diseases, such as peripheral artery disease and thrombosis. However, brain thromboembolisms, such as cerebral infarction, were separately classified as cerebrovascular diseases.

### 2.2. Statistical Analysis

Non-normally distributed continuous variables, as determined using the Kolmogorov–Smirnov tests, were reported as medians and interquartile ranges and compared using the Mann–Whitney U test. The effect size resenting the difference in the outcome data between the right and left side was calculated using the Hedge’s g. For comparing neck locations (lower, middle, upper), we used analysis of variance with post hoc corrections using the Kruskal–Wallis method. Categorical variables are reported as numbers and percentages, and they were compared using the χ2 or Fisher’s exact test, as appropriate. Continuous variables were changed to categorical variables to perform a statistical regression analysis on the factors affecting vessel area. Vessel area was divided into two groups (large and small) according to the median area. After division into groups according to the measured area at the middle level of the IJV, associations with demographic and clinical characteristics were analyzed using univariate logistic regression analysis. Variables were adjusted for age, sex, hypertension, cardiovascular disease, and body mass index (BMI). Large and small IJV areas were analyzed using multivariate logistic regression analysis, with the results reported as odds ratios (OR) and 95% confidence intervals (CI). Variables were tested for goodness-of-fit using the Hosmer-Lemeshow method. Two-sided *p*-values of <0.05 were considered statistically significant. All statistical analyses were performed using the SPSS version 21.0 for Windows (IBM; Armonk, NY, USA).

## 3. Results

During the study period, 743 patients underwent contrast-enhanced neck CT. A total of 429 patients were excluded from the study due to predetermined criteria, including 29 patients (6.8%) younger than 18 years, 132 neck operation patients (30.8%), 134 neck mass patients (31.2%), and 134 basic data missing (31.2%). There were 313 patients enrolled, including 118 with hypertension (37.7%), 63 with diabetes (20.1%), 44 with cardiovascular disease (14.1%), 75 with cerebrovascular disease (24%), and 70 with a history of chemotherapy (22.4%) (Figure 1). Among them, there were 160 males (51.1%), and 153 females (48.9%). The median age was 65.0 years, and the median BMI was 23.6 kg/m^2^. 

After comparing both IJVs, it was confirmed that the right side was larger than the left in all locations measured (Figure 2 and Figure 3). The median area of the upper IJV was 124.3 mm^2^ on the right and 89.1 mm^2^ on the left (*p* < 0.001, Hedges’ g 0.73). The median area of the middle IJV was 190.8 mm^2^ on the right and 127.0 mm^2^ on the left (*p* < 0.001, Hedges’ g 0.73). The median area of the lower IJV was 183.1 mm^2^ on the right and 94.5 mm^2^ on the left (*p* < 0.001, Hedges’ g 0.19) (Table 1). The diameter and area measured at the middle IJV of both sides were significantly larger than those at the upper or lower level (*p* < 0.001) (Table 2). The diameter and area of the right IJV at the middle level were found to be the largest. 

Since the middle level of the right IJV was the largest location in this study and is the most preferred location when accessing the central vein via the IJV, the regression analysis was performed with the right IJV (middle level) [10]. Table 3 shows the results of univariate analysis for the factors associated with the right IJV (middle level) area divided into two groups according to the median area (190.8 mm^2^). According to these results, cardiovascular disease (*p* = 0.003), age (*p* < 0.001), and BMI (*p* = 0.024) affected vessel size. All factors showing significance were further analyzed by multivariate analysis. Advanced age (OR 1.040; 95% CI 1.022–1.058, *p* < 0.001) and higher BMI (OR 1.080; 95% CI 1.011–1.154, *p* = 0.023) were factors significantly associated with the right IJV area (Table 4).

## 4. Discussion

The IJV catheterization is used in a variety of situations, including relatively short-term use, such as emergency or critical care, and long-term use, such as chemotherapy and hemodialysis [11]. Recently, it has become possible to safely and easily insert a catheter into an IJV by using ultrasound guidance [4,12]. However, in an emergency situation or in locations where there is no ultrasound machine, the procedure must proceed in a traditional method depending on external anatomical landmarks. IJV catheterization has no absolute contraindication, but it is cautioned against in certain situations, such as vascular damage, local infections, coagulopathies, and previous history of radiation therapy [13]. 

Attention must be paid during IJV catheterization using only anatomical indices as IJVs can have different anatomies depending on the patient [14,15,16]. In a study of hemato-oncology patients, 36% of them had unusual anatomical positions of their IJVs [15]. A study of healthy individuals reported that there was a difference in IJV area according to age, sex, handedness, and cervical spine level [14]. By contrast, our study found no difference in the right middle level IJV size with or without chemotherapy or by sex. The median right IJV (middle level) area was not different between those with chemotherapy and those without chemotherapy (190.87 (range 128.32–258.44) mm^2^ vs. 190.07 (range 137.74–256.38) mm^2^; *p* = 0.499) and also between male and female patients (196.94 (range 142.80–273.48) mm^2^ vs. 184.62 (range 132.70–241.46) mm^2^; *p* = 0.118). Rather, among the basic patient characteristics, there was a significant difference in the IJV area according to age and BMI. The older the patient, the lower the velocity of IJV blood flow and the lower the venous outflow volume [17]. Moreover, as the emptying of the atrium decreases, the venous blood pressure increases, which, following induction beyond the capacity of the jugular vein valve, leads to an increase in venous resistance and regurgitation [17]. These processes have been defined as venous dilatation. We could easily hypothesize that if a patient is clinically obese, then the thick fat layer could compress blood vessels [18]. However, in this study, we found that the higher the BMI, the larger the size of the IJV. Even though the mechanisms are unknown, it is known that a higher BMI results in higher abdominal and thoracic pressures, which can limit venous flow return and cause venous dilatation [14,18]. Although we were able to identify the factors that affected vessel size, we did not find clear cut-off values for practical application in the clinic.

The shape of the IJV is represented by a cylinder-like shape of a constant size in many medical anatomy textbooks [19,20,21]. Recent studies have shown that the morphology of the IJV has a conical shape that increases in size to the subclavian vein from the cranial vault [5,14]. Based on these results, it has been recommended to target the lower portion of the IJV when inserting a central line [5]. However, according to this study, the IJV was observed to have a rhomboid shape, which was larger at the middle level and became smaller above and below. Therefore, we suggest that there is no benefit in accessing the lower portion of the IJV, as it may unnecessarily cause complications such as pneumothorax during central line placement. In addition, the size of the right IJV was larger than the left at all levels. 

This study had several limitations. First, this study had a retrospective, single-center design and included relatively few patients. Care should be taken when interpreting and applying current results, as not all patients can provide accurate information given the retrospective nature of this study. Second, not all patients had the same physiological conditions. Depending on the patient’s condition, the size of the IJV may differ, but it was not possible to distinguish the patient’s condition at the time of taking the CT [22]. Third, it was not possible to determine the overall vessel state by measuring only partial vessel sizes without measuring all successive position levels of the IJV. Fourth, all underlying diseases that could affect the IJV size, including chronic cerebrospinal venous insufficiency, could not be identified. Therefore, a detailed analysis of the morphology of IJV may be necessary through the prospective study.

## 5. Conclusions

The right IJV is significantly larger than the left and has a rhomboid shape. Age and BMI were both significantly associated with blood vessel size. We believe that this result may be useful for choosing an ideal insertion location and reducing mechanical complications when accessing the IJV to insert a central vein line.

## Figures and Tables

**Figure 1 medicina-56-00135-f001:**
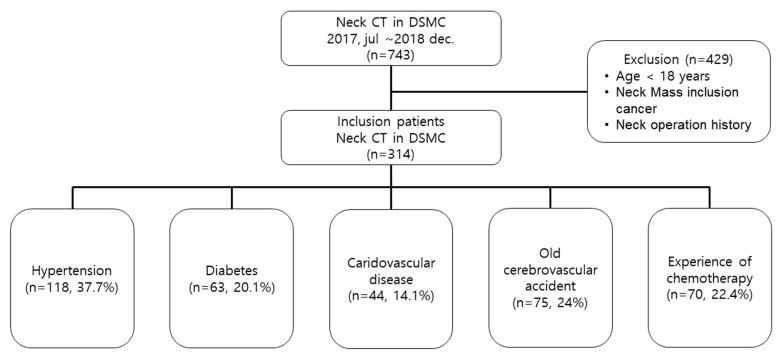
Flowchart of the study patients (n = number of patients, percentage in the enrolled patient group).

**Figure 2 medicina-56-00135-f002:**
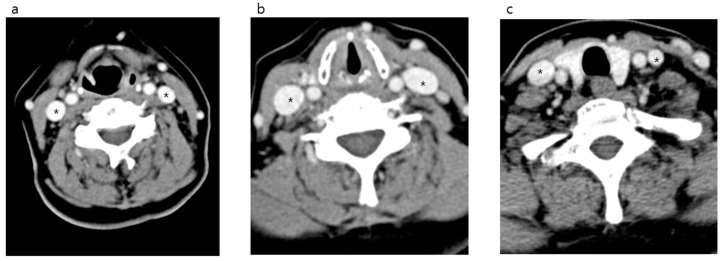
Neck computed tomography axial views of the internal jugular vein (*) and surrounding structures in the anterior neck. hyoid bone level (**a**). Cricoid cartilage level (**b**). First thoracic vertebra level (**c**).

**Figure 3 medicina-56-00135-f003:**
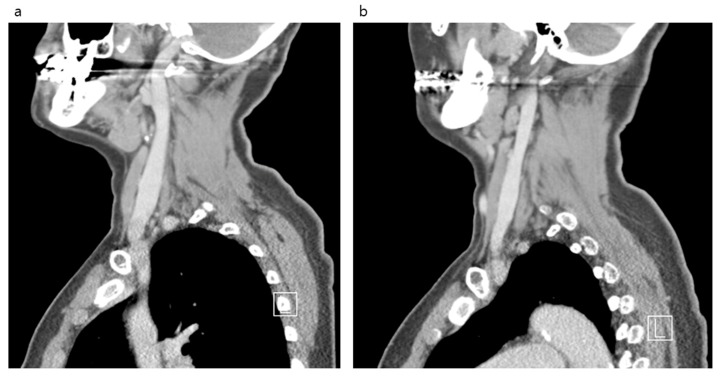
Neck computed tomography sagittal views of the internal jugular vein and surrounding structures in the anterior neck. Right side (**a**). Left side (**b**).

**Table 1 medicina-56-00135-t001:** Comparison internal jugular vein size by position level.

Location	Unit of Measure	Total	Right	Left	*p*-Value	Hedges’ g
Upper	Diameter (mm)	13.2 (10.9–15.7)	14.3 (12.3–16.8)	12.0 (10.2–14.6)	<0.001	0.69
	Area (mm^2^)	109.4 (76.5–153.2)	124.3 (98.7–176.2)	89.1 (65.1–124.9)	<0.001	0.73
Middle	Diameter (mm)	16.0 (13.0–19.9)	17.6 (14.8–20.9)	14.4 (11.8–17.5)	<0.001	0.73
	Area (mm^2^)	158.5 (105.8–226.7)	190.8 (137.2–258.0)	127.0 (88.0–180.2)	<0.001	0.73
Lower	Diameter (mm)	13.9 (11.4–17.3)	16.5 (14.2–19.1)	11.8 (10.2–13.8)	<0.001	0.77
	Area (mm^2^)	127.4 (89.0–196.3)	183.1 (127.6–241.6)	94.5 (68.7–126.7)	<0.001	0.19

**Table 2 medicina-56-00135-t002:** Comparison of internal jugular vein size by side of right and left.

Location	Unit of Measure	Upper	Middle	Lower	*p*-Value
Right	Diameter (mm)	14.3 (12.3–16.8)	17.6 (14.8–20.9)	16.5 (14.2–19.1)	<0.001
	Area (mm^2^)	124.3 (98.7–176.2)	190.8 (137.2–258.0)	183.1 (127.6–241.6)	<0.001
Left	Diameter (mm)	12.0 (10.2–14.6)	14.4 (11.8–17.5)	11.8 (10.2–13.8)	<0.001
	Area (mm^2^)	89.1 (65.1–124.9)	127.0 (88.0–180.2)	94.5 (68.7–126.7)	<0.001

**Table 3 medicina-56-00135-t003:** Univariate analysis of factors affecting internal jugular vein size.

	Total (n = 313)	Small (n = 157)	Large (n = 156)	*p*-Value
Sex, male, n (%)	160 (51.1)	75 (47.8)	85 (54.5)	0.235
Hypertension, n (%)	118 (37.7)	51 (32.5)	67 (42.9)	0.056
Diabetes, n (%)	63 (20.1)	30 (19.1)	33 (21.2)	0.652
CV disease, n (%)	44 (14.1)	13 (8.3)	31 (19.9)	0.003
Old CVA, n (%)	75 (24.0)	33 (21.0)	42 (26.9)	0.221
History of CTx, n (%)	70 (22.4)	35 (22.3)	35 (22.4)	0.976
Age (years)	65.0 (50.0-74.0)	58 (45.5-69)	69 (59–77)	<0.001
Height (cm)	161.0 (155.3–167.9)	162.0 (156.4–168.3)	160.1 (153.2–166.4)	0.099
Weight (kg)	62.0 (53.0–69.6)	61.9 (53.0–68.3)	62 (52.6–71)	0.358
Body mass index (kg/m^2^)	23.6 (21.2–26.3)	23.1 (20.8–25.5)	23.9 (21.6–27.1)	0.024

CVA: Cerebrovascular accident; CV: Cardiovascular; CTx: Chemotherapy.

**Table 4 medicina-56-00135-t004:** Multivariate logistic regression analysis of factors affecting internal jugular vein size.

	Odds Ratio	95% Confidence Interval	*p*-Value
Age	1.040	1.022–1.058	<0.001 *
Sex, Male	1.301	0.812–2.086	0.274
Cardiovascular disease	1.983	0.960–4.095	0.064
Hypertension	0.769	0.446–1.328	0.347
Body mass index	1.080	1.011–1.154	0.023 *

* mark means were factors significantly statistically associated with the right IJV area in multivariate analysis.

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
