# Peer review of "Anatomical Morphology Analysis of Internal Jugular Veins and Factors Affecting Internal Jugular Vein Size"

_medicina, 2020, doi:10.3390/medicina56030135_

Round 1

Reviewer 1 Report

I found this to be an interesting article, which makes a valuable contribution to the knowledge base. There is increasing evidence that the IJVs play an important role in regulating the fluid dynamics of the intracranial space and, as such, the results presented in the article provide a useful insight into the morphology of these vessels. Notwithstanding this, I feel that the article needs to be improved and strengthened in places (see comments below).

Detailed comments

1. The article does not state whether or not the subjects were supine when tested. I assume that they were supine and this should be stated in the methods section.

2. The cross-sectional area of the IJVs will alter throughout the cardiac cycle and also due to inspiration. How did the authors control for this in their study? A brief explanation of how this was accommodated needs to be included in the methods section.

3. The authors state: “Table 3 shows the results of univariate analysis of the factors associated with right IJV area divided into two groups according to their relation to the median value (190.8 mm2 ).” This classification criterion appears to be somewhat arbitrary, and there is no explanation in the text to justify its use. The authors should therefore add a justification of this methodological approach in the methods section. Also does the median value of 190.8 mm2 refer to the top, middle or bottom of the right IJV?

4. Logistic regression analysis was applied only to the right IJV. Why is this? Some justification for this decision needs to be included in the methods section.

5. The authors state that: “Age and BMI were both significantly associated with blood vessel size.” However, they do not state in which direction the association is. I assume that the IJVs got larger with increasing age and BMI, but this is not stated. Perhaps the authors could include the variable coefficients from the logistic regression mode, as these would give an indication as to whether or not the relationship was direct or inverse.

6. The authors’ state: “By contrast, our study found no difference in IJV size with or without chemotherapy or by sex.” However, the means and standard deviations of the IJVs are not presented for these respective groups. As such, this claim is unjustified. Therefore, the authors should include the IJV means and standard deviations for these respective groups in the text.

7. In Table 3 the specific statistical tests used should be indicated. Also, some explanation should be given as to what the figures in the brackets denote.

8. The limitations paragraph in the discussion section should be expanded to include a caveat about possible stenosis of the IJVs. If an IJV is stenosed at the distal end (as might be the case with CCSVI), then blood will tend to pool above the occluded section, cause the vessel to expand. If this is the case, then although an IJV might appear a good candidate for catheterisation, the vessel will lack patency and blood flow will be minimal.

Author Response

March 05, 2020

Dear reviewer:

I would like to re-submit an Original Article for publication in Medicina, titled “Anatomical morphology analysis of internal jugular veins and factors affecting internal jugular vein size.” The manuscript ID is medicina-737653.

The manuscript has been carefully rechecked and appropriate changes have been made in accordance with your and the reviewers’ suggestions. Changes have been marked in blue color font in the revised manuscript. The responses to the reviewers’ comments have been prepared and attached.

We thank you and the reviewers for your thoughtful suggestions and insights, which have enriched the manuscript. We hope that the revised manuscript is now suitable for publication in Medicina.

Thank you for your consideration. I look forward to hearing from you.

Sincerely,

Sang Hun Lee, M.D.

Department of Emergency Medicine, Dongsan Medical Center

Keimyung School of Medicine

1035, Dalgubeol-daero Dalseo-gu,

Daegu, Republic of Korea, 42601

Tel: +82-53-258-7895

Fax: +82-53-258-6305

Reviewer 2 Report

Jeon and colleagues characterised the anatomical size of IJV and found that age and BMI affected IJV size. While the study is interesting and may be of interest to clinicians, the significance of the study in alleviating clinical problems is unclear. Apart from suggesting how accessing the lower portion of the IJV may cause unnecessary complications, the author has not discussed how the results obtained can be used to further enhance the current clinical practice. The result section is very thin and analysis was superficial

Major:

Introduction needs more information to clearly show the clinical problem, preferably with stats. What kind of complications had been documented?

Please list out what cardiovascular diseases caused larger IJV.

I assume the IJV size was measured only at one position for table 3, when there was clearly a difference in size depending on the anatomical location of the patients. Was there any difference in IJV sizes at upper, middle and lower sections in each of the patients for the parameters analysed (BMI, age etc)? How does this compare to Table 3?

Was there any difference caused by the background (race) of the patients? What was the race composition of the patients?

Elaborate more on specifically how other clinicians should use this finding to choose an ideal insertion location of IJV? For example, “For older patients above the age of XX, we suggest XX part of the IJV” etc.

Minor:

Abstract: line 13/14 does not make sense.

Author Response

(The authors gave the same response as above.)
